# Wheelchair Dance: Exploring a Novel Approach to Enhance Wheelchair Skills, Belongingness and Inclusion among Children with Mobility Limitations

Jade Berthiaume [1,2], Claire Cherriere [3,4,5], Béatrice Ouellet [1,2], Laurence Éthier [6], Paula W. Rushton [7], Martin Lemay [3,8] and Krista L. Best [1,2,*]

1 School of Rehabilitation Sciences, Université Laval, Québec, QC G1V 0A6, Canada
2 Center for Interdisciplinary Research in Rehabilitation and Social Integration, Centre Intégré Universitaire de Santé et de Services Sociaux de la Capitale-Nationale, Québec, QC G1M 2S8, Canada
3 CHU Sainte-Justine Research Center, 5200, Rue Bélanger Est, Montréal, QC H1T 1C9, Canada; claire.cherriere.hsj@ssss.gouv.qc.ca (C.C.)
4 Toulouse NeuroImaging Center (ToNIC), Université de Toulouse, Inserm, UPS, 31024 Toulouse, France
5 Soins de Suite et de Réadaptation Pédiatrique, Paul Dottin ASEI, 31522 Ramonville St-Agne, France
6 Département de Danse, Université du Québec à Montréal, Montréal, QC H2L 1H4, Canada; laurence.ethier@gmail.com
7 School of Occupational Therapy, Faculty of Health, Dalhousie University, Halifax, NS B3H 4R2, Canada
8 Département des Sciences de L'activité Physique, Université du Québec à Montréal, Montréal, QC H2X 1Y4, Canada
* Correspondence: krista.best@fmed.ulaval.ca

**Abstract:** Playful approaches are recommended to enhance wheelchair skills training with young people. Inclusive dance allows participants to discover motor skills and improve social participation. Integrating wheelchair skills training into dance has not been evaluated. This study aimed to explore participants' experiences in dance while integrating wheelchair skills, and the influence of dance on wheelchair skills and wheelchair use confidence in young people. A convergent mixed-methods design was used during a one-week dance camp. Data collection combined observations, two focus groups (with young dancers who used manual wheelchairs and with professional dancers without disabilities), and evaluation of wheelchair skills and confidence. Data analyses included deductive thematic analysis guided by the Quality Parasport Participation Framework, merged with pre–post comparisons in wheelchair skills and confidence. Three young female dancers were 11, 12 and 15 years of age and three professional female dancers were 22, 27 and 27 years of age. Emergent themes included skill mastery, belongingness, and supportive environments. There were improvements in wheelchair skills and confidence (16.7%, 19.4%, 16.7%; 0.8%, 11.4%, 4.5%, respectively). Participants described overall positive experiences with the dance camp and perceived enhanced skills and confidence. This study advances knowledge about innovative approaches to integrate wheelchair skills training for young people. Future larger-scale controlled studies are needed to determine efficacy.

**Keywords:** dance; disability; inclusion; mixed-method; pediatric; training; wheelchair; young people

## 1. Introduction

Approximately 240 million children and adolescents (young people) have a disability worldwide and 7% of them have mobility limitations [1]. Mobility limitations restrict participation in activities promoting health, global development and quality of life [2,3], such as artistic and leisure-time physical activities. For children with disabilities aged 11–15 years, 8.5% to 40.4% meet the WHO physical activity guidelines of 60 min of moderate or vigorous intensity physical activity per day [4]. Young people with disabilities, and particularly young girls, participate in fewer leisure-time activities than their able-bodied

peers, engaging more commonly in sedentary leisure-time activities that involve fewer social interactions [5].

The importance for young people with disabilities to participate in leisure-time physical activity to their fullest potential has been highlighted by the UN Convention on Rights of Persons with Disabilities [6]. However, difficulties in performing fundamental movement skills has been identified as one of the most important factors explaining the low participation in these activities for young people with mobility limitations [7–9].

The use of a manual wheelchair allows young people with mobility limitations to move to places they want within their environment, which can increase participation in artistic and leisure-time physical activities. Being able to perform various manual wheelchair skills safely and independently can facilitate participation in these activities (e.g., rolling forward to catch a ball and turning in place or in a wheelie position to dance) [7–9]. However, many young people have difficulty performing such skills in a wheelchair and require training [10,11].

The Wheelchair Skills Training Program (WSTP) is a standardized program based on motor learning principles that can be used to teach indoor, community, and advanced manual wheelchair skills [12]. Numerous studies have demonstrated the effectiveness of the WSTP for improving wheelchair skills and confidence in adults, as well as satisfaction with participation in meaningful activities that require the use of a wheelchair [13]. However, the influence of the WSTP on manual wheelchair mobility in young people has only been explored in four studies with small samples ranging from one to eight participants [14–17]. Moreover, the program is underutilized in pediatric rehabilitation settings, thus limiting young people's opportunities to develop the manual wheelchair skills needed to participate in artistic and leisure-time physical activities [10].

Occupational therapists perceived that the use of a playful approach to train skills in young people could facilitate the use of WSTP in pediatrics [10]. Moreover, activity-based interventions in natural environment with an intense repetition respecting the child's goals and the 'just right challenge' by adjusting the task difficulty to the child's abilities are effective to promote motor learning in young people with disabilities [18]. Indeed, interventions for young people that are fun, promote contact with similar-aged peers with and without disabilities, are activity focused, and fitness oriented (physical activities) appear to be promising [19]. Among the activities recommended in pediatric motor rehabilitation that include these components, dance appears as a promising approach [20].

Dance is a popular physical and artistic activity that engages multiple motor and social skills for young people with disabilities [21,22]. For young people who use manual wheelchairs, participation in dance can be facilitated when they are able to perform a variety of skills with their wheelchairs [23]. Dance has many benefits, such as discovering motor possibilities [24], increasing self-esteem [25] and improving social participation [26]. Inclusive dance programs enable people with and without disabilities to dance together and promote interaction between dancers with and without disabilities through physical proximity, teamwork, and cooperation [27]. Since conducting dance studies exclusively on groups of people with a disability does not reflect their experience when they participate in leisure-time physical activities with people without disabilities and does not move toward a more inclusive society, more studies focusing on inclusive dance are needed [28].

Considering that playful approaches and activity-based interventions are recommended for the pediatric population and given limited evidence on innovative approaches for manual wheelchair skills training in pediatrics, an inclusive dance camp for youth integrating the WSTP was developed. The objectives of this study were to: (1) explore participation experiences of young people with disabilities and professional adult dancers without disabilities in an inclusive dance camp; (2) explore the influence of the inclusive dance camp on wheelchair skills and wheelchair use confidence in young people.

## 2. Materials and Methods

### 2.1. Design

A convergent mixed-methods design [29] including participative observations (ethnographic), semi-structured interviews (qualitative), questionnaires (quantitative) and clinical evaluation of wheelchair skills (quantitative) was conducted to explore the experience of participation and the influence of the dance camp training on participants' ability to perform wheelchair skills and confidence in using a wheelchair. Ethical approval was obtained from the Comité d'éthique et de la recherche du CHU Sainte-Justine (#MP-21-2015-827). Participants and their parents provided written informed assent and consent, respectively.

### 2.2. Conceptual Framework

Quality of participation, defined as a broad subjective evaluation of satisfaction, enjoyment, and perception of personally valued outcomes with a sport or leisure-time physical activity, can influence participation experiences [30]. In fact, quality of participation has been reported to have a stronger influence on well-being and social participation than increasing the frequency of physical activity alone [31]. The Quality Parasport Participation Framework (QPPF) facilitates the evaluation and improvement of participation experiences in adapted sports or leisure-time physical activities [30]; in this study, an inclusive wheelchair dance camp. The QPPF is comprised of six experiential elements explaining the quality of participation: autonomy (i.e., having independence, choice, control), belongingness (i.e., feeling included, accepted, respected), challenge (i.e., feeling appropriately tested), engagement (i.e., being in-the-moment, focused), mastery (i.e., feeling a sense of achievement and competence) and meaning (i.e., achievement of personal or socially meaningful goals). For example, in a wheelchair dance camp, participant autonomy can be promoted by allowing participants to choose music or dance movements, while challenge may be enhanced by choosing appropriate wheelchair skills that are required to perform a specific dance. Evans et al. [30] recommended the QPPF for use in exploring inclusive programs that include young people with and without disabilities.

### 2.3. Participants

The dance group included three young dancers (YD) who used manual wheelchairs and three professional adult dancers (PAD) without mobility limitations. The YD who used manual wheelchairs were recruited from the Marie Enfant Rehabilitation Center in Montreal (Canada) using a convenience sampling strategy. Occupational therapists shared invitation forms with young people who: were between 10 and 18 years old, used a manual wheelchair for more than 4 h a day, were able to propel their manual wheelchair independently for a distance of at least 10 m, and understood simple instructions with a two-step sequence (determined by the referring clinician). Young people with medical contraindications to physical activity practice, who received surgery within the past 6 months, or had a moderate to severe intellectual disability were excluded. PADs without mobility limitations were recruited using a purposive sampling strategy. The dance teachers of the camp invited PADs from their network who were at least 18 years of age and presented good interpersonal relationship skills (e.g., attentive, caring, funny, reliable).

### 2.4. Dance Camp

*Dance camp structure.* The dance camp lasted five consecutive days (three days at the Marie Enfant Rehabilitation Center and 2 days in a dance studio), six hours per day with a 2 h session every morning, a 1 h break for lunch and a 3 h session every afternoon (i.e., total of 30 h). Morning sessions were dedicated to the acquisition of dance knowledge focusing on space, rhythmic structure, quality of movement, relationship with the other dancers and learning of short choreographies integrating the WSTP skills. Choreographies integrated opportunities to adapt movements according to each YD's skill level (e.g., to roll forward as usual or in wheelie position). During afternoon sessions, participants were accompanied in the process of creating a choreography with an inclusive perspective. Each YD who

used a manual wheelchair was paired with one PAD as recommended by Burridge and Nielsen [32]. A live performance and video of the dance choreography were presented in the dance studio to the participants' families and professionals of the rehabilitation center at the end of the camp. This event was preceded by a meeting with a professional dancer with mobility limitations using a wheelchair in order to stimulate their engagement, as recommended by Morris, Baldeon [23].

*Dance camp team*. The dance camp was facilitated by two dance teachers, one bachelor of occupational therapy student and two occupational therapists specialized in wheelchair skills training (one doctoral student with clinical expertise in pediatrics and one researcher). The two dance teachers were always present and rotated within each session, with one leading while standing and the other assisting while using a wheelchair, as recommended by Morris, Baldeon [23]. Both dance teachers have been trained in teaching inclusive dance (one as a physiotherapist with training in adapted and inclusive dance and the other as a school dance teacher). They also received a 3 h wheelchair skills training with one of the occupational therapists to be able to dance in a wheelchair using the proper techniques. A filmmaker and a musician were invited to create the music and a video of the choreography to stimulate the creative process and to allow dancers to collaborate with guest artists [23].

*Daily post-camp meetings*. Meetings were held every evening post-camp to adjust the program and activities according to participants' energy levels, needs, preferences and responses to the activities that were carried out during the day. As the dance camp progressed through the week, meetings were helpful to determine ways to better incorporate wheelchair skills training in the dance activities.

*Integration of the wheelchair skills training*. During the first two days of the camp, wheelchair skills training consisted of group practice of skills based on motor learning principles with only YDs. Starting on day three until the end of the camp, skills training was integrated throughout dance or musical games involving all dancers to facilitate group cohesion, engagement, and motivation. During the dance and musical games, the adult dancers were invited to experience using a wheelchair and the YDs who used manual wheelchairs gave them wheelchair skills training tips. An example of one game was the musical train, in which everyone followed the first person who set the rhythm (i.e., the conductor), and when the music stopped, the conductor had to perform a challenging manual wheelchair skill and everyone else had to perform the same skill after. The bachelor student and the two occupational therapists helped the morning and afternoon dance sessions when required by teaching the techniques to perform wheelchair skills using motor learning principles. The principles employed included demonstrations in combination with verbal instructions, provision of extrinsic (e.g., verbal knowledge of performance) and intrinsic (e.g., using bubble wrap [auditory feedback], a mirror [visual feedback]) feedback, segmentation, simplification, and progression. Wheelchair skills training activities were integrated in the morning or afternoon dance sessions depending on the energy level of the dancers.

### 2.5. Data Collection

*Observations*. Participant observations were conducted throughout the camp by the occupational therapy student (JB). Documentation of individual factors, interactions with other participants and camp facilitators were collected in a reflexive journal. Some sessions were filmed, and photos were taken to be analyzed and included in the observations. A daily logbook completed by the two dance teachers prior to and at the end of each camp day also supported the documentation of the contents, the process, and the adaptation of the activities of the dance camp.

*Interviews*. One post-dance camp focus group was conducted by the occupational therapy student (JB) with YDs (60 min) and the PADs (90 min). The purpose of each focus group was to explore participants' perceptions, experiences, and quality of participation in the inclusive dance camp. The semi-structured interview guide was the same for both the children and the adults and explored the six experiential elements of the QPPF influencing

the quality of participation [30]. Ten open-ended questions were used to facilitate the discussion (e.g., "How does the gathering of people with and without mobility limitations to dance make a difference for you?"). Focus groups were audio-recorded.

*Questionnaires*. Physical activity enjoyment and flow were, respectively, measured using the French versions of the Physical Activity Enjoyment Scale (PACES) questionnaire [33] and of the Flow State Scale (FSS) questionnaire (short version) [34]. These questionnaires have been used in a dance program for children with mobility limitations [35] and the PACES was validated in pediatrics [33]. The PACES asks participants to rate level of agreement with 16 items about activity enjoyment on a 5-point Likert scale (1 = "Strongly Disagree" to 5 = "Strongly Agree"). Each level of agreement was converted to a percentage score (where 1 = 0%, 2 = 25%, 3 = 50%, 4 = 75%, 5 = 100%) and a mean percentage score was calculated. Scores above 50% are indicative of agreement with enjoying the activity, while scores below 50% are indicative of disagreement with enjoying the activity. The short version of the FSS comprises 17 items measuring the perception of the flow in physical activity [34]. Flow is described as a psychological state where the challenges of a situation and one's personal skills are matched, characterized by a pleasurable feeling of engagement in the activity. The participant answers with a 5-point Likert scale (1 = "Strongly disagree) to 5 = "Strongly agree"). Each level of agreement was converted to a percentage score (where 1 = 0%, 2 = 25%, 3 = 50%, 4 = 75%, 5 = 100%) and a mean percentage score was calculated. Scores above 50% are indicative of agreement with experiencing flow during the activity, while scores below 50% are indicative of disagreement with experiencing flow during activity.

The Wheelchair Use Confidence Scale for Children Using a Manual Wheelchair, French version (WheelCon-M-F-P) was administered to the YDs pre- and post-camp. The WheelCon-M-F-P is a self-report questionnaire in which children rate their self-efficacy level on a 5-point ordinal scale for 33 items that represent different situations of daily life using a manual wheelchair in six areas: negotiating the physical environment, performing activities in the manual wheelchair, knowledge and problem solving, advocacy, managing social situations, and managing emotions. Self-efficacy is the construct evaluated in the WheelCon-M-F-P. For simplicity of understanding, the word 'confidence' will be used rather than 'self-efficacy'. To score the WheelCon-M-F-P, each response was first transformed to a number (red face = 0, tan face = 1, yellow face = 2, light green face = 3, green face = 4) and then the average questionnaire total score was calculated by adding the scores of each question and dividing it by the number of questions answered. The questionnaire total score was transformed into percentages by dividing it by four (maximum possible score) and then multiplying it by one hundred as follows: ((sum of each item score)/([number of items] × 4)) × 100. The WheelCon-M-F has been validated in a sample of 22 children [36].

*Clinical test*. Wheelchair skills capacity of YDs was evaluated using the Wheelchair Skills Test (WST) Version 5.2 Form. Among the possible 32 skills from the WST, the following 12 skills most likely to be integrated and trained in the dance camp were selected and evaluated by direct observation prior to and after the dance camp by the occupational therapy student (JB) (Table 1). The total wheelchair skills capacity scores were calculated using the following equation: Total WST Capacity Score = sum of individual skill scores/([number of possible skills − number of No Part scores − number of Testing Error scores] × 3) × 100. The WST has been validated in the adult population (good test–retest reliability; excellent intra- and inter-rater validity) [12] and in one study with a small group of twelve children with spina bifida [37]. Every YD using a manual wheelchair also chose two skills from the WST to improve during the dance camp as individual goals. This selection was performed prior to the administration of the WST to avoid a training effect [12].

**Table 1.** The 12 selected skills assessed with the Wheelchair Skills Test (WST) and their descriptors.

| # | Skills Selected | Descriptors |
|---|---|---|
| 1 | Rolls forward | YDs self-propel the wheelchair forward on a smooth level surface over a 10 m distance. |
| 2 | Rolls backward | YDs self-propel the wheelchair backward on a smooth level surface over a 5 m distance. |
| 3 | Turns in place | YDs turn the wheelchair around in right and left directions in as tight a space as possible. |
| 4 | Turns while moving forward | YDs turn the wheelchair to the left and right while moving forward. |
| 5 | Turns while moving forward | YDs turn the wheelchair to the left and right while moving backward. |
| 6 | Maneuvers sideways | YDs maneuver or "shuffle" the wheelchair sideways to the left and right using as little forward–backward space as possible (e.g., similar movement to parallel parking with a car). |
| 7 | Reaches objects | YDs pick an object up from the floor and also grab an overhead object (1.5 m high from the floor). In dance, similar up and down movements are performed without picking objects. |
| 9 | Shifts weight | YDs which relieve weight from the sitting surfaces on both sides (i.e., lateral trunk flexion on right and left sides). |
| 12 | Performs ground transfers | YDs move from the wheelchair to the ground and back. |
| 26 | Performs stationnary wheelie | YDs maintain balance on the rear wheels over a 30 s period. In dance, the position can be held for a shorter time. |
| 27 | Turns in place in wheelie position | YDs turn the wheelchair to the left and right in the wheelie position. |
| 28 | Rolls forward and backward in wheelie position | YDs move the wheelchair forward in the wheelie position on a smooth level surface over a 1.5 m distance. The movement is then performed backward. |

YD: young dancer.

*Procedure.* For the YDs, the WheelCon-M-F-P, the selection of two skills from the WST as individual goals and the WST were (in order) administered pre- and post-camp. The PACES, FSS and focus group were conducted three days after the end of the dance camp. For the three PADs, PACES and FSS were completed right after the end of the camp whereas the focus group was completed the next day.

### 2.6. Data Analysis

The sociodemographic data were analyzed using descriptive statistics (e.g., median, interquartile range (IQR), frequency, percentage) The PACES and FSS results (median (IQR) score) were summarized. The WheelCon-M-F and WST results (individual score for each item and individual total score) were calculated for every dancer using a wheelchair prior to and after the dance camp. Interview recordings were anonymized and transcribed verbatim using TranscribeMe, and then transcripts were verified by a team member. A mixed approach to data analysis was used. First, deductive thematic analyses (QPPF) were performed using NVivo 1.7.1 software to organize the data. Data were coded using the six experiential aspects of QPPF and the foundation to these elements consisting of physical, program and social environments that are safe, welcoming and inclusive [30,38]. This analysis was performed by (JB) and verified by another member of the research team (CC). Information was triangulated using observations, questionnaires, and clinical test scores [29]. Integration of the qualitative and quantitative data is presented through merging and comparison in the result section. Observations provided more context on participants' perceptions, whereas questionnaire scores supported participants' statements by providing a concrete measure of their perceptions regarding their participation and abilities. The clinical test score provided an objective measure of the change in their wheelchair skills [29].

### 3. Results

*3.1. Participants*

Three YDs and three PADs participated in the dance camp. Table 2 presents sociodemographic information of the dancers. Two YDs were schooled at home, and one was schooled at a rehabilitation center. Two PADs had experience in teaching young people with disabilities. Table 3 presents the WST skills that were selected as individual goals by the YDs.

**Table 2.** Participants characteristics.

| Participant | Gender | Age | Diagnosis | Number of Years of Experience Using a Wheelchair | Experience in Dance |
|---|---|---|---|---|---|
| YD 1 | F | 12 | Neuromuscular disease (Transverse myelitis) | 1 | Yes |
| YD 2 | F | 15 | Spinal cord injury (Paraplegia) | 4.5 | No |
| YD 3 | F | 11 | Neuromuscular disease (Spinal muscular atrophy type 2) | 8 | Yes |
| PAD 1 | F | 22 | N/A | N/A | Yes, for 24 years |
| PAD 2 | F | 27 | N/A | N/A | Yes, for 4 years |
| PAD 3 | F | 27 | N/A | N/A | Yes, for 19 years |

YD: young dancer; PAD: professional adult dancer.

**Table 3.** Two skills from the Wheelchair Skills Test (WST) selected by each young dancer (YD) as individual goals.

| Participant | Skill Selected as Individual Goals |
|---|---|
| YD 1 | Reaches objects |
| | Rolls forward and backward in wheelie position |
| YD 2 | Turns in place |
| | Reaches objects |
| YD 3 | Maneuvers sideways |
| | Reaches objects |

*3.2. Experience of Participation in the Dance Camp and Influence on Wheelchair Mobility*

Table 4 presents a summary of the WST, WheelCon-M-F-P, PACES and FSS scores for each participant. Participants' perceptions regarding their participation in the dance camp are presented according to the six experiential elements of the QPPF framework and the safe, welcoming and inclusive environment that support the experiential elements (objective 1). Belongingness and meaning thematic were combined, as participants' statements very often related to both dimensions. Results on wheelchair mobility outcomes (capacity, confidence, and individual goals) are integrated into the QPPF elements to which they relate, thus complementing qualitative data (objective 2).

**Table 4.** Summary of the WST, WheelCon-M-F-P, PACES and FSS scores (%) for each participant.

| Participant | WST Total Score (Pre) | WST Total Score (Post) | WheelCon-M-F-P (Pre) | WheelCon-M-F-P (Post) | PACES | FSS |
|---|---|---|---|---|---|---|
| YD 1 | 44.4 | 61.1 | 65.2 | 65.9 | 89.1 | 65.4 |
| YD 2 | 38.9 | 58.3 | 72.7 | 84.1 | 81.2 | 76.9 |
| YD 3 | 72.2 | 88.9 | 76.5 | 81.1 | 93.8 | 96.2 |
| PAD 1 | N/A | N/A | N/A | N/A | 100 | 69.2 |
| PAD 2 | N/A | N/A | N/A | N/A | 89.1 | 82.7 |
| PAD 3 | N/A | N/A | N/A | N/A | 93.8 | 76.9 |

PACES: Physical Activity Enjoyment Scale; FSS: Flow State Scale.

*3.3. Mastery*

YDs expressed enjoyment learning new techniques to perform manual wheelchair skills, such as placing casters in a trailing forward position before leaning to reach an object on the floor: "the little wheels, that's something new I'd never heard before. [...] And that allowed me to learn it [...] I found it interesting. [YD 3]". The three YDs reported improvements in their manual wheelchair mobility during the camp, for example in doing transient tips with their wheelchairs (i.e., lifting the front casters slightly off the ground). Furthermore, mastering different techniques allowed them to not only use wheelchair skills in dance settings, but also to integrate them into their daily lives. YD 3 explained, "the lateral movement, I didn't really use it before. Now, I learned how to do the movement [...] When I move, I tell myself I can do a lateral movement to achieve such thing". The improvements were also apparent in the WST results with individual scores increasing by 16.7% (YD1), 19.4% (YD2), 16.7% (YD3) from pre- to post-camp.

Among the wheelchair skills selected by each YD as individual goals, the skill "Turns in place" improved by 66.7% (YD2), the skill "Maneuvers sideways" improved by 33.3% (YD3) and there were no changes in the four other skills.

Both YDs and PADs reported that YDs had a higher confidence level in using their wheelchairs: "I have improved my confidence with my wheelchair [YD1]" and "you could see the girls gaining confidence in their artistic expression, depending on their ability [PAD 2]". YD 1 also mentioned that the wheelchair skills training activities were her favorite and that all the activities at the camp helped her to improve her confidence in using her wheelchair. The level of confidence remained almost the same for YD1 (0.8%) and improved for YD2 (11.4%) and YD3 (4.5%) from pre-to post-camp.

PADs also noticed improvements among YDs. Observations made by PAD3 on YDs' capacity to perform the transient tip illustrate this evolution: "just like doing little pops [...] they had us do the exercise with the bubble wrap on the floor and everything; she got it, like, on the first try, whereas at the very beginning, it was more challenging". That the PADs tried the different wheelchair skills themselves helped them to notice improvement and mastery among YDs. It was observed that the YDs were encouraging PADs to try the skills and were proud to teach them the techniques. In fact, PAD1 noticed: "They were the ones who showed us, who taught us a little bit about how things were done, [...] it motivated them. So, the roles would change during the week".

Additionally, peer exchanges and peer learning were observed throughout the camp. For example, YD1 and YD2, who were less experienced, were watching YD3's movements and trying to imitate her in the stationary wheelie practice.

Finally, dancers mentioned that they could benefit from a camp with a longer duration to master the wheelchair skills they practiced, to learn new ones to integrate into their choreography and to reduce stress associated with the rapid learning of choreography and wheelchair skills to be able to perform a video clip and a show: "I found that it was very busy and short. So, it brought in some stress that could have been avoided; either having a two-week camp or lower expectations". [PAD 2] and to develop skills: "[...] more time to do the creation, more days, to work more with your partner, things like that [YD3]." [Video clip is available at https://www.youtube.com/watch?v=vHPecfOkdcg accessed on 20 November 2023].

*3.4. Belongingness and Meaning*

YDs stated that it was rare for them to be around other adolescents in wheelchairs and reported this as a benefit of the camp. However, YD3 mentioned: "The atmosphere was good, but I think the age difference [between young people and adults] made a bit of a clash". Indeed, the three YDs would have liked to have had moments of exchange independent of the rest of the group: "I think we all have things in common. We can share everything. It's like not every day that you're going to be like this with people [...] who are in wheelchairs too. Yeah, well for once we could have talked about it" [YD1]. It was also observed that the YDs did not interact frequently with each other by themselves until the

last day of the camp. PAD3 specified that the camp is a rich opportunity for teen dancers to meet: "These are young people who are quite isolated regarding their mobility. Then, it recreates a social circle. [. . .] Even at the end, I saw YD3, she was like, 'Come on, YD1, let's go practice the wheelie' [. . .] a new friendship was created just like that" as she noticed interactions on the last day. Participants also stated that being around other wheelchair dancers (teachers and YDs) enhanced their belief in their possibilities: "It also inspires me to see everything that's possible because it seems to me . . . that there's not much that's so possible for me, when on the contrary, sometimes, it can even be an advantage [YD3]". The possibility that the PADs could move on the YDs' wheelchair to perform acrobatics while dancing together was perceived as an advantage by YD2, and YD1 specified: "This was beautiful".

Regarding interactions with the PADs, YD3 explained: "There was a great atmosphere. But yeah, maybe, two weeks, we might have had time to dance more with others, develop more bonds". However, YD1 reported changes regarding her level of comfort with people entering her own bubble. Indeed, she discovered she was less comfortable than what she thought with having unknown people close to her in the beginning of the camp: "Sometimes I wasn't too comfortable when people touched me. I wasn't always comfortable touching other people". Additionally, YD1 identified that she was shy because she was unfamiliar with dance partners, "I found over time, I started [. . .] to be less uncomfortable when people touched me [. . .] They are new people, so we needed to get to know each other". In this regard, the three YDs appreciated that PADs asked for their consent before interacting and dancing with them or touching their wheelchairs.

### 3.5. Challenge

Participants considered the level of challenge to be medium, which they appreciated as YD1 stated, "I wanted it to be challenging, but [. . .] I wanted something that was relaxed, that was fun, and then you were learning things. And still with some effort".

For the adult dancers, some mentioned that they could have had a higher level of difficulty, especially in the movements where they were not interacting with the YDs. However, PAD2 specified that the context was not oriented on performance according to her: "I don't think I pushed myself technically, but I don't know if I would want to push myself technically, in a context like that. I find that the objective is more in the meeting with the other" [PAD2].

### 3.6. Engagement

Different strategies were used to stimulate engagement in the program. For example, using a playful approach integrating dancing to the practice of wheelchair skills was particularly appreciated by all dancers, as explained by YD1: "That's what I preferred. [. . .] When we put on music [. . .] that said, like: «Jump» Well, you had to lift up your casters". In fact, the game to practice the transient tip using bubble wrap and music (i.e., propelling the wheelchair at a speed following the rhythm of the music and, in front of each strip of bubble wrap placed on the floor, lifting casters) was one of the YDs' favorite activities.

PAD1 also observed an increased engagement among all participants when the structure of wheelchair training activities was modified around the middle of the camp to use a more playful approach and include everyone: "I found it very well structured: the classes, the workshop times, and even more since it was worked on in game mode, that we got so involved, all week long of course, but I found that when there was music as from the middle of the week (i.e., when music was added in the specific wheelchair training times during the dance sessions) [. . .] it motivated them even more". As such, dancers noticed the adjustments to programs and activities that were made along the camp.

Participants strongly agreed with experiencing flow during (median FSS score of 76.9%) and with enjoying (median PACES score of 91.4%) the inclusive dance camp, which may facilitate feelings of engagement. Moreover, YDs experienced a good balance between soft and calm music and more energetic rhythms used in the choreography which they all

enjoyed. PAD3 expressed how inspiring and motivating the meeting with a professional wheelchair dancer and artist was: "[. . .] it was a major highlight of the week's program, I think, because having feedback like this from an adult in the same situation, who is just as able as anyone else of wanting to dance, and of making a living out of something that motivates her, I find it very inspiring. And I hope that it can motivate them, [YDs] and give them the desire to continue".

### 3.7. Autonomy

Even though the studio was an environment really appreciated by dancers, independent access to the building was limited. Wheelchair users needed help to enter the building and to open the doors of the elevator to access the studio. YD1 mentioned: "It was just annoying that, like, you were by yourself, and then you couldn't really take it (the elevator)".

Generally, in the dance camp, participants felt they were free to choose to participate in an activity or not: "there are always activities proposed, but there was never an obligation to do them. It's a proposal, it's an invitation, you give what you can give [PAD2]".

Finally, involvement in the creative process, namely in the choice of music and creation of the choreography, was perceived as engaging and brought pride to dancers, as PAD1 and YD3 said: "In the creation, they [the teenagers] have been active in everything they have done regarding the music (. . .)" [PAD 1]. "I preferred the creation [. . .] I am used to do choreographies that are already planned. I find that it adds like a, a little bit of pride when we are the ones creating it" [YD3]. Musical propositions were showed at the beginning of the camp by the musician according to YDs' musical preferences. After the first presentation of the music, it was observed that YD3 specified what kind of music she would want for her number, and the musical arrangement was then modified by the musician to better fit her preferences. YD2 also mentioned she appreciated integrating her favorite movement in her creation.

### 3.8. Safe, Welcoming and Inclusive Environment

Dance camp structure. The structure was appreciated and was adjusted over the five days to the needs and energy of the dancers: "I thought the structure was really, really good; to start in the morning, afterwards, to see according to the girls' energy for the week. [. . .] Let's say, it's like, one morning, we started early with the creation. So, that was good, because that way, the energy was concentrated in the morning". [PAD 1]. However, integration of wheelchair skills training into dance in a welcoming, fun and inclusive way was a difficult task and was modified throughout the camp after daily debriefing. There was kind of a 'clash' between the dance activities and the wheelchair skills training. Therapists felt they were disturbing the activities when teaching a skill or giving tips while the YDs were dancing, and they did not want to separate participants with disabilities from the group to practice skills individually. These findings prompted the team to reflect during the daily post-camp meetings and propose a new approach from day 3, where the training was brought into playful training activities integrating dance movement, music, collaboration with PADs and everyone's creativity. Along with training activities the skills were integrated in dance choreography. For example, the skill 'reaches objects' was practiced with fluid movement of the hands going to the floor and back. All participants mentioned that they enjoyed this new format and that it better met their needs.

Number of participants. The number of participants was found to be appropriate according to the PADs: "For me, [. . .] I found that we really were the right number [three teenagers and three adults]. More, in the space we had, with everyone's bodies, with the wheelchairs and all that, it would have been really complex. Then, even to make connections, really, all together, not to lose the concentration, then the energy of everyone, I find that it is good". [PAD 1].

Inclusive environment. YD3 explained she was often infantilized in her daily life because of the height difference between an adult standing and a youth sitting in a wheelchair,

and she often felt the need to prove herself. However, the inclusive environment helped her: "Eventually, I realized that I didn't have a point to prove here. But it's like a reflex that I end up taking, when I meet new people". Adult dancers explained how dance, to its very essence, promotes inclusion in their relationship and interaction with others: PAD 1: "when you dance or when you work with any person, it's a different reality. [. . .] There's a communication done through the body, which is even more present, but all that, it's something I could really relate just through our practice and not just through disability".

Dance camp location. The transition from the rehabilitation center to the dance studio had several benefits according to the participants. First, YD3 mentioned that she preferred the space that a studio offered: "For sure, whenever I'm in a studio, I really like it because I really have like a lot of space to do a lot of stuff". In addition, all participants mentioned preferring the studio for the brightness. Also, YD1 and YD3 agreed on the preference for a place outside of hospital settings: "Even if you say like, you know, for once I come here [to the rehabilitation center] like to have fun, me what I was thinking is like I'm coming back to the same place for like the fourth time this week [YD3]".

## 4. Discussion

This study explored the participation experience of YDs and PADs in the dance camp and the influence of the dance camp integrating the WSTP on the wheelchair skills and wheelchair use confidence in young people. Participants expressed a positive experience of participation in the camp and demonstrated improvements between pre- and post-camp with respect to wheelchair skill capacity and wheelchair confidence. Challenges were faced such as the lack of harmony between the dance activities and the wheelchair skills training or the age gap between PADs and YDs. Several facilitators supported engagement and positive experience such as the development of knowledge and competency on wheelchair skills by all the participants (mastery), social support created by the group (meaning and belongingness) and access to a safe and inclusive physical environment. The discussion is presented according to these three key themes from the QPPF framework that emerged from this study to explain the experiences of participation of all participants and the influence of the dance camp integrating wheelchair skills training on YDs' wheelchair skills and confidence in using their wheelchair.

### 4.1. Mastery

Although the dance camp was only five days in duration, improvements in individual WST scores from pre- to post-camp were observed in wheelchair skills capacity, even for the YDs who had more years of wheelchair experience. Wheelchairs skills training was distributed throughout the 5 days of dance camp which may appear as a short period to improve wheelchair skills. YDs explained they would have preferred to extend the dance camp to two weeks to improve and develop their skills. The team chose to conduct an intensive dance camp because several studies suggest that intensive rehabilitative approaches (with intense repetitions) are associated with improved neuroplasticity and global motor function over the long term [18,39,40]. Even though the most optimal duration and frequency remain unknown for young people, some research suggests that condensed training may not be the most suitable option to promote skills development in this population and that, consequently, frequent and shorter training sessions distributed over a longer period of time appear preferable [12,15,17]. In particular, for advanced skills that have had no improvement in WST results (e.g., perform a stationary wheelie) along with the complexity of the dance activity integrating wheelchair skills, to allow more time for rest between training sessions and consolidation or to vary contexts in which wheelchair skills are practiced could be beneficial to improve dance camp [15,17]. Further studies would be needed to determine whether intensive therapy is more effective than conventional therapy when interventions are distributed over a longer period of time [41].

For wheelchair skills that had no improvement in WST results (e.g., perform a stationary wheelie and roll forward in a wheelie position), qualitative improvements were noticed

(e.g., better control of timing when executing a transient tip, time increased while rolling forward maintaining a wheelie position). The qualitative results are therefore necessary and relevant to gain a better understanding of the influence of the camp on wheelchair skills capacity. Among the individual goals selected by YDs, two YDs improved one of their goals. This improvement could be related to them being more engaged and focused as they noticed real progress towards their own goal [42,43]. The goals that have maintained the same score post-camp could be explained by a lack of understanding of the task demand (insufficient knowledge about the skills at the beginning) which may have led to the establishment of unrealistic goals for a one-week camp. The dance camp may have had an influence on other movement skill dimensions not captured by the WST as dance allowed to practice different movements in the space (e.g., spatial awareness while navigating in the space with a partner or a group) and time (e.g., moving with a control of timing according to the music rhythm). The study of other movement skill dimensions was outside the scope of this study, but they could be developed within a dance context for evaluation in future studies.

*Motor learning principles.* Wheelchair skills were practiced and integrated creatively in the choreography created and learned by participants. During wheelchair skills training activities especially, trainers used different sources of extrinsic (e.g., verbal knowledge of performance) and intrinsic feedback (e.g., visual and auditive feedback with bubble wrap) adjusted to the learning stage of each YD to support motor learning. Moreover, motor learning principles suggested by the WSTP [12] were successfully integrated into the dance camp: integrating a routine in the dance camp, individualizing the dance process into subgroups of two dancers (PAD supporting YD in reaching its goals), offering demonstration with a dance teacher in a wheelchair or with a YD directly to other YDs and breaking the skill down into its component for some more advanced skills (e.g., stationary wheelie skill broken down into sub-steps).

*Playful approach for wheelchair skills training in dance.* YDs particularly enjoyed these activities as they were fun and helped them to develop their skills. Enjoyment, satisfaction, and well-being of all participants during dance camp was also supported by the PACES and FSS positive scores. Learning through playful and meaningful activities was a successful approach which supported YDs engagement as it is recommended in a growing number of studies [10,44,45]. Integration of wheelchair skills training into dance had some challenges the team had to overcome and was modified throughout the camp to a playful approach merging training activities with dancing games, combining meaningful activities for participants in the camp (i.e., play and dance). The Inclusive Dance Program presented in the Morris and al. [23] article also came to similar conclusions about the playful aspect of dance and the importance of less structured exercises in which the goal was still to learn movement-related skills to support their engagement as they explore their own creativity through practice. Interventions in pediatric rehabilitation are more effective when the intervention is activity-based ('top-down' interventions) and the occupation, meaningful to the child [18,46]. Play is considered as one of the most important occupations for children. Playful approaches in pediatric rehabilitation are associated with many benefits such as a better engagement in therapy as the child experience pleasure while developing his skills, along with development of problem-solving skills as they challenge themselves, better inclusion and opportunities to socialize [46,47].

*Confidence.* Findings in this current study suggest a positive effect of the dance camp on the YDs' confidence in performing wheelchair skills and to integrate them in their everyday life to overcome barriers in their environment (e.g., maneuvers sideways in a tight space). This finding is consistent with findings from Bloemen, Backx [8], whereby participation in adapted sports programs showed positive influence on children's confidence level and how child's confidence is positively associated with engagement in physical activity and improvement in their mobility and their independence. In fact, confidence is a determining factor predicting the adoption of physically active behaviors over time [8,48].

### 4.2. Belongingness and Meaning

*Modeling*. Mastery of the skills also appeared in modeling and learning by observation throughout the dance camp as YDs exchanged tips and tricks and were encouraging each other to try new skills. YDs explained how seeing other people in wheelchair executing movement showed them new possibilities with their wheelchair and increased their confidence in trying them out. In fact, peer-training can increase physical activity participation of children with and without disabilities, increase peer interaction, improve confidence in one's ability to learn new skills and generate behavior changes (e.g., increase wheelchair mobility) [17,49]. A peer trainer may also be considered more credible than other professionals, as they have lived experiences using their wheelchair [50].

Appropriate schools, sport clubs and community opportunities are important facilitator factors to support peer-training and participation in leisure-time physical activities for young people with disabilities [8]. In this study, as two YDs were schooled at home, and one was schooled at a rehabilitation center, they could also benefit from more opportunities to interact with others experiencing and sharing similar aspects in their lives. Leisure-time activities like dance camps provide important opportunities to promote social interaction between young people using manual wheelchairs. Age differences between PADs and YDs appeared to be an obstacle to the feeling of belongingness among the YDs, who would have preferred to have moments of exchange and social interaction among themselves.

Different strategies proposed by YDs could also be integrated in the dance camp to further support peer-training. The strategies proposed included scheduling activities where YDs could be grouped together to provide opportunities to talk and dance together, and to integrate same-aged peers without disabilities. It was also suggested that increasing the duration of the camp may encourage prolonged exchanges that could facilitate the development of meaningful longer-term interpersonal relationships between participants.

### 4.3. Access to Safe, Welcoming and Inclusive Environment

*Inclusive social environment*. The dance camp also created unique opportunities to test their capacities in safe and supportive environments. This inclusive aspect also allowed nonwheelchair users (e.g., instructors, adult dancers) to explore movements in a wheelchair as a new way to encourage collaboration between participants and to share a unique experience between all participants. Research team members ensured that everyone (YDs and PADs) was comfortable and consented to the activity beforehand. The teacher model (e.g., YD during all participants' exploration in a wheelchair) and learner (e.g., nonwheelchair users when they explored movement in a wheelchair) roles were then exchangeable throughout the dance camp between YDs, PADs, trainers and dance instructors depending on type of activity. Dance exploration using a wheelchair by nonwheelchair users encourages collaboration and revisit notion of distinctions between able and disabled bodies. New dance techniques can also be discovered and empower everyone involved in the project [23].

*Inclusive and meaningful physical environment*. It appeared important to YDs to practice dance in the real environment in which this activity usually takes place (i.e., studio) instead of being in a rehabilitation center. Even if there were existing physical barriers to access the studio (e.g., physical assistance to enter the building), YDs preferred this location. A similar impression emerged from Goodwin, Krohn [27] who sought to understand the wheelchair dance experiences of children, aged between 6 and 14 years, with spina bifida. Participants expressed how the dance studio environment supported their engagement in dance as they felt totally at ease and felt it was 'their space'.

### 4.4. Study Limitations

The small sample size limits generalizability and the results should be interpreted with caution. However, the findings provide useful insight for future inquiry about inclusive wheelchair dance camps as a potential way to enhance wheelchair skills and quality of participation. An additional limitation is that one member of the research team conducted

all observations during the dance camp and facilitated the focus group. While this may have helped to establish rapport and trust with participants, facilitating collection rich information about lived experiences, the constant presence of this researcher during the camp may have led to social desirability bias. For example, participants may have been less comfortable sharing negative aspects about the camp. Furthermore, the researcher may have been less objective while conducting the focus group. Given the focus group was conducted to gather an in-depth understanding of participants' experiences, this presented a limitation for full inclusion of YD2 in this component of the data collection due to her social and communication difficulties. Individual interviews may be considered in future studies.

Given a sub-set of 12 wheelchair skills were evaluated (chosen by researchers as the skills most likely to be integrated and trained in the dance camp), the reliability and validity of results may have been affected. Finally, the dance camp required considerable human resources (ten adults, including the PADs) for three YDs which may represent economic challenges for reproducibility. However, the intention of this study was not to evaluate efficacy or cost-effectiveness, but instead explored how wheelchair skills training could be integrated into an inclusive dance camp, and the quality of participant on all dancers. Given this was the first study exploring wheelchair skills training in dance, additional human resources were necessary to ensure safety and to include those with the expertise required to adapt the training approaches as the dance camp progressed. Future studies may only require one wheelchair skills trainer and would not necessitate professional dancers.

## 5. Conclusions

Participants had an overall positive experience at the dance camp. Results suggested a positive influence of the dance camp integrating wheelchair skills training on Yds' capacity to perform wheelchair skills and confidence in using the wheelchair in their everyday life. The positive findings in this exploratory study are promising and support the potential of playful, activity-based approaches to manual wheelchair training in pediatrics. This study offers a better understanding of the elements promoting or limiting a positive experience for young people in an inclusive dance camp incorporating WSTP. Future larger-scale studies would validate the results obtained in this exploratory study, so that this innovative wheelchair training approach can be used with greater confidence in pediatrics.

**Author Contributions:** Conceptualization, C.C., M.L., B.O., L.É., P.W.R. and K.L.B.; methodology, C.C., M.L., B.O., P.W.R. and K.L.B.; data collection, J.B., C.C., L.É., B.O. and P.W.R.; data analysis and interpretation, J.B., C.C., B.O., P.W.R., K.L.B. and M.L.; writing, J.B., C.C. and B.O.; review and editing the manuscript, J.B., C.C., B.O., L.É., P.W.R., M.L. and K.L.B. All authors have read and agreed to the published version of the manuscript.

**Funding:** Financial support was provided for this research through the Hewitt Foundation. Salary support was provided to Jade Berthiaume by the Cirris and the Université Laval Faculty of Medicine (Summer Research Scholarship—Faculty of Medicine), Claire Cherriere (Hewitt Foundation), Beatrice Ouellet (Quebec Health Research Funds, Canadian Institutes of Health Research Vanier Scholar, Participation Sociale Ville Inclusive; Ordre des ergothérapeutes du Québec), Paula W. Rushton (Quebec Health Research Funds Junior 2 Scholar Award) and Krista Best (Quebec Health Research Funds Junior 1 Scholar Award).

**Institutional Review Board Statement:** This study was conducted in accordance with the Declaration of Helsinki, and approved by the Ethics Committee of CHU Sainte-Justine (#MP-21-2015-827).

**Informed Consent Statement:** Informed consent was obtained from all professional adult dancers and parents whose children were involved in this study. Assent was obtained from all young dancers who participated in this study.

**Data Availability Statement:** The data presented in this study are available on request from the corresponding author due to privacy and ethical reasons.

**Acknowledgments:** The authors would like to thank the professional adult dancers and young dancers who participated in this study, Marion Chuniaud-Lacau for directing and editing the video clip and Sergio d'Isanto for the original musical composition.

**Conflicts of Interest:** The authors declare no conflicts of interest. The Hewitt Foundation was not involved in the conceptualization and conduct of the study, in the collection, analysis and interpretation of data and writing of the manuscript.

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
