# Peer review of "Wheelchair Dance: Exploring a Novel Approach to Enhance Wheelchair Skills, Belongingness and Inclusion among Children with Mobility Limitations"

_disabilities, doi:10.3390/disabilities4010014_

Round 1

Reviewer 1 Report

Comments and Suggestions for Authors

Wheelchair dance : A novel approach to enhance wheelchair skills, belongingness and inclusion among children with mobility limitations

General:

The article covers an interesting topic with clinical relevance. The findings could help promote a new method of WST combined with leisure in an ecological approach.

The articles would benefit by condensing the content. Many sections and quotes are wordy. Maybe because they were translated from French. It would also benefit from editing the English as some sentences are very long and difficult to understand,

The YD seem to all be experienced wheelchair users. The chronic nature of this study should be emphasised: even people who have been in using a wheelchair for most of their lives, can improve in a week camp! That is quite amazing.

Title: The title should include that it is an exploratory study.

Abstract:

Line 18: recommended to enhance wheelchair skills training – no need for “implementation”.

Line 21: add that it was an exploratory study.

Line 27: age range or ages of the professional dancers should be included as they were also studied (and the age difference between YD and PDA was found to be important)

Line 29: The improvement in wheelchair confident should be moderated. 0.8% is an improvement, but only slight.

Introduction:

Lines 50-52: “Among the numerous factors explaining the low participation in these activities for young people with mobility limitations, difficulties to perform fundamental movement skills has been identified as one of the most important”.  Try writing it in the opposite order: Difficulties to perform fundamental movement skills has been identified as one of the most important factors explaining the low participation in these activities for young people with mobility limitations. This is just one example. The article would benefit from some linguistic editing to make it easier to read.

Line 69: ranging from one to eight participants(?)

Line 102: The second goal needs to be clarified. It seems to be to examine the influence of the dance camp on wheelchair skills and confidence. It says that the aim is to explore the influence of integrating WST in a camp, but we have no way of knowing whether the improvement was specifically due to the WST integration rather than the whole camp.

Materials and methods

Concept of the framework – It is important paragraph, but too long. Try to shorten.

Dance camp structure: This paragraph is very interesting.

Team: Was it an issue that for 3 YD there were 3 PAD+ 2 teachers + 3 Ots +1 filmmaker +1 musician (10 adults)? The economic challenge of this ratio needs to be discussed in limitations.

Line 256: WST: It is unclear if the 32 skills were evaluated or only the 12 skills in the appendix.

Line 262: The skills chosen as individual goals. They are not mentioned later in the article. It would be interesting to know which skills were chosen (perhaps add it to the table in the appendix) as well as to know if these skills improved more than other skills. This would be interesting in the discussion and could relate to motor learning as those skills are more meaningful and salient.

Table 1: The number of years the participant used a wheelchair would be interesting (instead of mobility limitation)

Experience of participation: please shorten the quotes. Also, there is repetition that can be reduced. 

Line 309: maybe skills instead of “tricks”?

Line 321: the improvements are even more impressive if you emphasise that the YD all used wheelchairs for many years, but managed to improve in a one week dance camp.

Line 329: A 0.8% improvement does not seem very significant. There should be a qualifier.

Line 324: The sentence order should be reversed.

Line 351-2: The link to the video clip is great!!

Line 357: The age difference should be discussed in the discussion.

The paragraph about the level of difficulty for the PAD could be shortened as the article is more about the YD.

Line 422: It is not quite clear what 76.9% on the FSS means. Does a score above 50% mean enjoyment?

Paragraph starting line 423: It isn’t clear to me how the type of music and the teacher being in a wheelchair fits in with enjoyment (PACE).

The goals the YD chose does not seem to be presented in the results.

Discussion:

Line 503: maybe leave out “accomplished”. Try something like: This study explored the participation experience…..

Line 538: WST without change are not mentioned in results. It would be interesting to see a table of the change in each skill for the participants, including which skills were chosen as goals. The goals are not mentioned in the discussion, which seems a pity as it is interesting to understand if salience would motivate and a larger change be seen.

Line 586: Are you trying to say that you hope the YD will continue to be physically active due to the increase in confidence. Or, it is important that they feel confident as this is predictive of physical activity. Try to make it clearer, please.

Lines 609-614: Please divide this sentence. It is hard to follow such a long sentence.

Study limitations

There might be a limitation that the researcher who did the focus groups was at the whole camp and was less objective. Also, this might cause bias as the participants won’t feel comfortable saying negative aspects.

The small number of participants should be mentioned as well as the high staff requirements that make it less economical.

It is a great study, and I hope you make these small changes as it will contribute greatly to the body of knowledge as well as clinical practice. 

Comments on the Quality of English Language

Some sentences are too long, making it hard to follow. Also, some sentences need re-writing. Sometimes the order of the sentence should be reversed. I assume they are used to French and that effects the writing. 

Author Response

We have provided our responses to Reviewer 1 in the attached file. 

Reviewer 2 Report

Comments and Suggestions for Authors

Purpose and methods of the research are well-articulated by the authors. I understand that the authors chose to incorporate the quantitative measures with the qualitative outcomes in the results section, however, I suggest that a table of results summarizing the PACES, FSS, WheelConM-F-P, and WST(12 skills) outcomes be included in the results section, as a useful point of reference for the reader. Specifically because the article is addressing "wheelchair skills," it is important that the 12 skills in Appendix A that were assessed are actually discussed, clearly. Some of them are touched upon within the narrative description, others are not.

For the wheelchair skills, please explain "maneuvers sideways" and "shifts weight."

Page 11, paragraph 3, line 538, describes some of the qualitative improvements in wheelchair skills that occurred through the dance program, such as better control of timing -- this is a significant insight, and a suggestion is to develop this area of the discussion further, perhaps identifying other wheelchair skills or movement skill dimensions not captured by the WST, but that could be unique and usefully developed through a dance context. For instance, if dancers worked in partners and groupings, moving through space with their wheelchairs, perhaps spatial awareness and the ability to navigate space more skillfully was enhanced within the dance context? How might this translate to better navigation of obstacles and people in daily living environments? Spatial and rhythmic/timing skills are commonly trained in dance environments -- if these aspects were a part of the dance program in this research, how might these dance-based skills impact the young students' management of their bodies and their assistive devices? 

A limitation of the study is the very small sample -- 3 dancers using wheelchairs and 3 nondisabled adult dancers, but this is not mentioned in the limitations section.

Comments on the Quality of English Language

Quality of English language is pretty good throughout the article. However, there are several areas that need improvement to aid clarity.

Page 11, Discussion section, line 510, use of the word "lived"?

Page 11, line 516, missing word, "of"

Page 11, line 536, interventions

Page 6, line 281, remove period

Page 12, line 597, helped

Page 12, line 606, should the word "less" be more? 

Page 12, line 607, consist of

Page 12, line 608, wheelchairs

Page 13, top paragraph needs to be more concise and clearer, re-phrase for clarity

Page 13, line 618, recommended change: The inclusive aspect also allowed nonwheelchair users...

Page 13, line 622, consented

Page 13, line 626, re-phrase for clarity

Page 13, line 632, associated with

Page 13, line 638, delete "as a sense of belongingness" or re-phrase

Page 13, Study limitations, re-phrase this section for better clarity. Check verb tense. 

Author Response

We have provided our responses to Reviewer 2 in the attached file. 
